# Religious Influences on the Experience of Family Carers of People with Dementia in a British Pakistani Muslim Community

**DOI:** 10.3390/healthcare11010120

**Published:** 2022-12-30

**Authors:** Peter Kevern, Dawn Lawrence, Nargis Nazir, Anna Tsaroucha

**Affiliations:** School of Health and Social Care, Staffordshire University, Stoke-on-Trent, Staffordshire ST4 2DE, UK

**Keywords:** Muslim, dementia, carer, religion, British Pakistani, BAME, service provision

## Abstract

This paper reports on a study that begins to address the paucity of research around the religious motivations of Muslim carers of family members with dementia. Seven carers were recruited for interviews from the British Pakistani Muslim community concentrated in the Midlands and North of England. Interview transcripts were analysed thematically using an iterative collaborative methodology. The findings suggested that the Muslim faith plays a pivotal role as a support mechanism for individual carers and their families, but the wider faith community and its leaders did not typically offer support and could impede access to external care. This was a result of cultural pressure and lack of awareness both among religious leaders and the community as a whole. The study concluded that the inequality in access to dementia services may be constructively addressed if service providers engage with these faith concerns in the community and religious leaders to meet the needs of Muslims of British Pakistani origin.

## 1. Introduction

There are currently at least 700,000 informal carers for people with dementia in Britain, and caregivers from Black, Asian and Minority Ethnic (BAME) communities are substantially more likely to provide family care than their White British counterparts [1]. Most BAME communities in the UK suffer from reduced access to and uptake of a range of health and social care services intended to support people with dementia and their carers, as noted in a series of studies [2,3,4,5,6,7]. Among communities of people of South Asian origin, there is a pattern of late diagnosis and low uptake of support services affecting both those living with dementia and their carers.

The research focus in terms of ethnic inequalities in access to dementia care is on ‘culture’, and any reference to religious influences tends to be treated as one aspect of ethnic difference. However, for most communities of South Asian origin in the UK, including those of British Pakistani origin, religious beliefs, practices and organisations have an important influence both on their experience and on their response to community challenges: for this reason, these religious influences deserve research in their own right. The responses of Christian communities and individuals to dementia have received some attention, as have those of Sikhs. However, much less is known of the role of religious influences on dementia care in Muslim communities. The purpose of this paper is therefore to explore the significance and main features of the religious dimension of the carer experience in the case of a group that has tended to be hard to reach and resistant to such research: Muslims of British Pakistani origin (This term is intended to include Muslims who were born in Pakistan, as well as those who left North India at the time of Partition rather than migrate to Pakistan. In includes Muslims of this community who were born in the UK. It is not intended to include those from the British Bangladeshi community, or Muslim communities originating elsewhere in the South Asian region. For simplicity, we will use the term ‘Muslims of British Pakistani origin’ throughout the paper). The aim is to provide initial insights into the role of religious beliefs, practices and community networks on dementia care and identify the ways in which the Muslim religion influences the understanding of, response to and access to care for people living with dementia in this community. The findings will prepare the way for a larger scale study to inform future strategies for targeted support among Muslim communities of British Pakistani origin.

## 2. Background

Among communities of South Asian origin, there has been a significant amount of research pointing to a complex of reasons for late and reluctant access to dementia services, including family expectations, ignorance of dementia, economic concerns, cultural values and culturally inappropriate services (e.g., [8,9,10,11,12,13,14,15,16]. In addition, there is some evidence of differences in the response of service providers themselves when presented with a service user from a BAME community, including those of South Asian origin [17]. These findings have led to calls for more culturally sensitive services [18] s and even for separate services for minority ethnic groups [19].

Some more recent research has tended to focus on how ‘cultural values’ influence dementia help-seeking in particular subgroups such as the British Bangladeshi community in East London [20]. However, despite the fact that many communities of South Asian origin show high levels of religious adherence compared to the White British population, much less work has been done on how religious norms, beliefs and practices in turn influence, stabilise and encode those ‘cultural values’ or how understanding of the religious dimension may inform future attempts to address inequalities of care in minority groups [21]. At the level of belief and practice and their significance for dementia generally, there is a growing body of literature relating to Christianity [22,23,24,25]; and there are a few high-quality studies relating to the Sikh community [26,27,28,29]. Extending the search outside the UK, there are two studies of Jewish communities from Israel and the USA [30,31] and a study on Buddhist belief and practice among carers of people with dementia in the USA [32].

There is a gap in the research base concerning Muslim beliefs and practice in relation to dementia care. There is a little literature from South Africa [33,34], a few papers from Pakistan [35,36] and a perceptive study of Arab Muslim caregivers in the Middle East [37]. Finally, there are two recent attempts to understand dementia from a specifically Islamic perspective [38,39] which indicate the possibility of an international dialogue emerging in the foreseeable future.

In the UK context, Muslims of South Asian origin represent the second largest religious grouping after White British Christians, and one in which the number of people living at home with dementia is predicted to rise rapidly [2]. Nevertheless, in this context, only three papers have been published, exploring the beliefs of Muslim family carers of Bangladeshi origin [20,40,41], along with one that identifies changes in their understanding of dementia in response to the rise of revivalist Muslim beliefs [42]. One may infer a similar relationship between religion and caring in the community of Muslims of British Pakistani origin which predominates in the Midlands and North of England, but no research has yet been conducted in this community. Furthermore, there are reported differences between the cultural expressions of Islam prevailing in Bangladeshi and Pakistani immigrant communities, and between different communities of Pakistani origin [43,44] which raises questions of the transferability of these findings. Finally, at 2% of the UK population the British Pakistani community is substantially larger than the British Bangladeshi population (at 0.8%) and, after the British Indian community, the second largest non-White community in the UK [45]. It therefore merits further research in order to establish how Muslim belief and practice is supporting, or impeding, the endeavours to ensure appropriate dementia care for this underprovided group.

## 3. Materials and Methods

### 3.1. Methods

A qualitative scoping methodology was employed as the most appropriate method to fulfil the study purpose. Scoping studies are a widely used method used to gain an overview of a particular topic area, with the aim of informing future research, policy and practice. The study used semi structured interviews to explore participants’ views, practices and experiences in caring for a relative with dementia with particular focus on the religious aspects of these views, practices and experiences. 

Ethical approval was obtained from Staffordshire University Research Ethics Committee (date reference 7 June 2021). 

### 3.2. Participants

An opportunity sample of seven participants with an age range between 25 to 60 and an average age of 43 years, who care for a relative living with dementia was recruited. There were four female and three male participants; three cared for their grandparent and four for their parent, whilst one was a carer for both parent and spouse. This was the maximum number that could be obtained from the community in question and is towards the bottom end of sample sizes for a qualitative study (Braun and Clarke suggest 6–10 participants are needed for a small study [46]) but was adequate for the purposes of scoping the topic. Data saturation was not sought, as the purpose of the study at this stage was to identify themes for further research. 

The eligibility inclusion criteria for participants were: Muslim, from British Pakistani background, who have or had experience caring for a relative with dementia at home, able to communicate in English without an interpreter. 

All recruitment was undertaken in May and June 2021. It was conducted online via virtual networks or by telephone due to the COVID-19 restrictions current at the time. Participants were recruited in the first instance using the contacts of one of the researchers, and then by a snowballing technique. This sampling technique is particularly appropriate for hidden populations which tend to be difficult for researchers to access. In this instance Muslim women’s online social networks provided participants who then suggested friends and contacts of their own who may be willing to participate. Finally, a general message was circulated through a WhatsApp group for Muslim Women. 

Once a potential participant had agreed to give their contact details, they were contacted by email or post with further information and a consent form for the interview. If they agreed, the information sheet and consent form were sent by email directly to the researcher, who checked conformity to the inclusion criteria and arranged a date for the interview. Written informed consent was obtained prior to interview via email or post and verbal consent was also obtained at the point of recording the interview.

### 3.3. Data Collection

An interview schedule was developed by the team to guide the interview process based on the aims of the study as described earlier. Interviews took place in English between 13 July and 10 September 2021. Because data collection took place under COVID-19 restrictions, participants were interviewed online or via the telephone and an audio recording was made which was uploaded to the Otter.ai software platform. Recordings of interviews were auto-transcribed using the Otter.ai platform and then manually checked for transcribing errors by each of the four researchers separately. Words that were difficult to understand on the recording or were in a language other than English were discussed by the team and a consensus interpretation was agreed. The agreed transcripts were then used for the thematic analysis

### 3.4. Data Analysis

Thematic analysis was undertaken using the six-stage approach articulated by Braun and Clarke [46,47]. The first stage of analysis (Familiarization with the data) was conducted in consultation: one member of the team produced a spreadsheet of extracts from each transcript that had apparent reference to religious beliefs, practice or community involvement and this was then revised in team discussion. The next steps (Generating initial codes; Searching for themes) were undertaken by each of the four researchers independently. Individual code lists with initial themes arising from them were then compared and the remainder of the process (Reviewing themes; Defining and naming themes; Producing the report) was undertaken collectively through a process of iterative discussion and revision. 

### 3.5. Reflexivity

All interviews were conducted by a single member of the research team who was a female Muslim of British Pakistani origin and was therefore likely to be more acceptable to the interviewees from both a linguistic and cultural point of view. The research team encompassed ethnic, religious and gender diversity which enabled different perspectives to be brought to the planning and analysis stages of the project and comprised of four members: a female British Pakistani Muslim; a female Muslim Revert of Greek origin (The term revert is used by some Muslims rather than convert based on the belief that everybody is born as a Muslim (i.e., submitted to God), brought up in different religions and then returns (reverts) to Islam); a female Black British Christian of Caribbean origin; and a male White British Christian. All are based at a University in the West Midlands region of the UK.

## 4. Results

Initial coding and the production of proposed themes was conducted independently by the four authors. Collation of the proposed themes and discussion of the underlying evidential base in the coding process led to an agreed structure of nine themes classifiable into three categories which, with hindsight, corresponded to the three levels of social organisation shaping the experience of family carers (see Table 1). These themes are discussed in turn below.

### 4.1. Carers’ Religious Beliefs and Practice

#### 4.1.1. Providing Motivation to Care

Carers practiced their religion to different degrees, but all referred to their religion as important to them and to the person living with dementia. Most of them spoke of the religious injunction to care for parents as providing them with motivation and an expectation they sought to fulfil (Here and throughout, extracts are given verbatim to allow the authentic voice of the interviewees to come through. Colloquialisms, repetitions and partial sentences are therefore present in the extracts given. Where a word from another language is used, the English equivalent is given in square brackets).

So yes, I would say religion is the main thing is it teaches us to, you know, look after our elders and, all the values that we get are from religion my, you know, most important thing is religion.P3

I think, as Muslims, it’s very clear… that we look after our elders, and we should provide care and support and look for our elders, and know the Quran says don’t say *Uff* to your parents to your elders. There’s a push that we take that responsibility, we, as I said that we provide a loving, caring environment, we look after them and support them.P7

for me, is very, very important, very important. I think everything I do is, is around my religion. So being kind to my neighbors, doing my prayers, fasting, celebrating spending time with my family, being patient with the elders and the link to the younger…P2

#### 4.1.2. Providing Strength and Support in Caring

In addition, a number referred to their practices of faith and/or prayer as sources of strength, patience and release from anxiety, so enabling them to fulfil their family duties/responsibilities. This is broadly in line with a number of studies showing personal religious and spiritual practice as reducing caregiver burden [48].

You find solace in your faith and your religion. And I definitely did find peace in it. And I suppose you know, when you read certain verses, when especially the one, particularly one from the hadith, “After much hardship comes peace” really resonated with me. Because we went through, my mom went through much hardship, and we did as well. *alHamdulillah* [praise be to God], you know, you come out a better person.P6

I think religion plays a big role in your thinking around some coping strategies that you know what, what helps you what makes you not lose your temper? I think it’s natural human to get angry at times when you get frustrated. Caring is really difficult, can be quite lonely, isolating.P7

I didn’t think about financial implications, how I’m going to sustain myself and pay the bills. But I left it to [Allah’s will] Allah wants from me, He will do the best for me and my family.P5

Furthermore, several interviewees spoke of their religious practice as building their strength or supporting the character traits that helped them to care faithfully:

So in Islam, we are always it’s wise to be patient. … You have to, you know, have a good faith and, you know, good values. And, you know, look after someone with dementia, because you need a lot of patience. And you know that I’m a Muslim and in Islam patience is a key. So that has helped us a lot as well.P3

I would say like, if you’re fasting, you get, I don’t know from where but you get a lot of patience and you know, it hasn’t made it difficult for us now.P3

It [faith] made me actually stronger. It gives me satisfaction in front of myself and my family. And I also tried to be to the Almighty, that he should accept my duty, effort.P4

Finally, some interviewees spoke of the hope of a divine reward as a source of support in fulfilling their family duties:

It’s a bit logically thinking it’s just it’s like we do we do we do an activity to gain from it. And as Muslims, we don’t always get it from in this world and we’ll get it from the next one. So it’s a bit of like, when I look after my mom it’s like an opportunity that’s been given to me to gain rewards, so yeah, definitely.P2

So, it has made me realize that, you know, when you learn about Islam more, you learn that, you know, if I’m caring for her, in future, something good is gonna come to me. You know, gonna come to me. P3 that was one of the reasons why I wanted to look after as well and it was, it was also in the hope of, as I’m a firm believer, of, you know, being rewarded for that, if not in this life, then you know, the next.P5

#### 4.1.3. Influenced by the Experience of Caring

Since the practice of Islam stresses faithful observance of particular times and seasons–for daily prayers and for the annual fast of Ramadan–we anticipated that the demands of caring for somebody living with dementia would prove disruptive for religious practice. There was some evidence for this:

So it can have a massive impact [on religious practice]. I can imagine my mom’s sort of in between me doing my prayer, depending how embedded it is and how it impacts their memory, that they will try and disturb me. We’re halfway to reading my prayer.P2

I’d probably say that it’s difficult. You know, sometimes when, like, you know, it’s the time to pray and you have to provide, like, you have to put them on priority. So if they are hungry have to make them food first so I wouldn’t say that it has affected a lot, but sometimes it can be…P3

However, some carers took a pragmatic approach, regarding their expectations both for the person living with dementia and for themselves:

I think they just managed it [Ramadan] as you would normally in a family. If she got up, she got up, you know, there’s no, there’s no push. She was old as well, there’s no push for her to have to get up to fast or anything. So if she did wake up, you know, she could be part of the family. She didn’t wake up, that’s fine. Let her sleep. And it was the same during the day, when it was time for her to eat we would make her some food, but we didn’t think oh well, should she try fasting or anything, because, you know, it wasn’t obligatory for her and the family just continued as normal.P7

Sometimes I would be late [with prayers] because, you know, sometimes an emergency would happen that could be mum had an accident she would have an incontinence accident. … the priority then is to make sure, mum’s okay. So I know it may not sound, you know, acceptable or whatever, but at that time when you when you in that moment and you are caring for someone dependent … everything else doesn’t matter.P5

In addition, the religious practice or perspective of the person living with dementia sometimes proved to positively support the religiosity of the carers:

And I remember it was Ramadan at the time. And I was really struggling… And I remember nanny was saying to remember nanny was saying that you don’t need to worry about that. … you’re forgiven, because you’re, you’re caring for somebody who can’t care for themselves as somebody who’s got has got no body. … Really kind of, you know, really kind of molded the way I became as a person in terms of my religion as well, they brought me closer, definitely.P5

Yeah, I mean, my mom would do [read *namaaz*, prayers] always [read *namaaz*] always having [*tasbi*, prayer rope] in her hand, at the end of her life, we would have *talawat* [recitation of the Quran] on a CD or something like that all the time. So religion plays a massive part. So I wouldn’t say we … it probably strengthened it [my faith].P5

To summarise the findings in this section, the analysis indicated that the British Pakistani Muslim carers in our sample attributed many positive elements of the experience of caring to their religious beliefs and practices. These provided motivation to accept the caring task; and incentives, values and practices that support the caring task. 

These findings suggest that support for British Muslim Pakistani carers of family members living with dementia will yield better fruit if it works with their religious beliefs and practices than if it ignores them. They raise the question of how carers’ faith is worked out in practice in relation to the wider Muslim community of which they are a part.

### 4.2. Muslim Community Response to Dementia

In contrast to the positive and flexible account of individual practice, carers generally reported their communities as rigid in their expectations of carers and their duty as faithful Muslims. At the root of this, several respondents cited a confusion between the demands of culture and religion as their reason for determining how dementia is perceived within the Muslim community.

#### 4.2.1. Culture/Religion Distinction

As has been frequently noted in the religious studies literature, the distinction between cultural and religious beliefs and practices is not clearly perceived or understood in stable communities with a single dominant religion, and reflection on the difference between religion and its cultural expression may only take place when encountering a different culture [49]. This distinction will therefore be more dominant in the younger generation and distinguish their approach from their elders’ [50].

In practice, interviewees tended to weave concepts of culture and religion together to varying degrees. Thus, for some, the two appear almost interchangeable:

… it’s built within us, whether it’s our social values, it’s our cultural values, if it’s our ethical kind of like values, they often come out of there. And so whilst you might not overtly say that I’m doing this because I’m a Muslim, or I’m doing it because, God says so, I think you, you do that without thinking because your faith is important to you, you know that your faith is going to look after them.P7

religious is… guide you to take care of your parents, and also that in our culture this is encouraging we must look up to our parents at that old age.P4

However, some were critical of this confusion as overlooking the needs of the carers themselves:

… as Pakistani heritage family, it’s almost like you don’t challenge your elders and your elders are able to get away with it. So, if you, for example, when one couldn’t manage. … for mum the carer said that for me, as a daughter in law, as a Muslim, as a Pakistani [inaudible] my job is to just get on with it, and try and do the best that I can to support her, rather than try to find a solution out of the house. … They expect you to care and provide that care, because that’s your duty. They’re not interested in how you are coping.P7

but culturally, there’s an expectation [in the community] what will the community say so what would your neighbour say? That man there is abandoning his parents because people conflate and we’ve said this before, people conflate nursing accommodation, a care home and a nursing home all together but they’re very, very different.P5

The cultural expectations function as a deterrent to family carers seeking outside help, and in particular external residential care. As will be further discussed below, Islamic teaching need not be expressed in this way.

#### 4.2.2. Awareness and Understanding of Dementia

Previous studies have drawn attention to the relatively low levels of awareness, knowledge and understanding of dementia in communities of South Asian origin living in the UK, and this state of affairs was borne out in the present study by all of the respondents:

Other people are not aware of some of these illnesses. My father was diagnosed in 1999, the majority of the people were not aware of this illness. So, the people, people didn’t understand even [when it was] explained them.P4

However, some awareness and understanding of dementia is starting to emerge in the community:

And also as I’m learning more about the condition about especially with faith, we talked about this before, and the fact that, you know, the Imam talked about verses from the Quran, and I was aware of that link directly to the fact that dementia is a disease of the brain. And this is something that was revealed 1400 years ago, only science is only now realizing what this is. So that to me is really powerful. And I use that a lot when I speak with other carers.P6

#### 4.2.3. Community as a Barrier or Potential Source of Support 

Understandably, misunderstanding and ‘misdiagnosis’ are a feature of the community’s lack of knowledge of dementia, and lead to barriers to the provision of support by the wider British Pakistani Muslim community. Clearly a key issue here is that the lack of knowledge of dementia in the community means it is not engaged with the issues or alert to the need to offer support as part of their faith:

I think there’s not a lot that can be said on that, because there isn’t a lot of awareness [of dementia in the community]. So without, without the words, people don’t appreciate or realize the difficulties that you might be going through what might term as the burden of care, people, people don’t necessarily see that. They expect you to care and provide that care, because that’s your duty. Not interested in how you’re coping, they’re not interested in whether you’re caring at home or you’re trying to, you know, the words will be like making excuses not to care.P7

I think Muslim community needs a bit of more education on this all because it is quite new subject for them as well. … So now, like, you know, we feel a bit of support from them, but we thought it was hard, you know, explaining it to everyone. … Okay. And then they, they take time to understand, okay, and then they show their support to us, but not not that much.P3

We go to *Masjids* [Mosques] and listen to the [inaudible]. But we don’t hear about how, when we say, we should visit someone when they’re ill, make sure they’re okay, how often do we do that as Muslims? Do we, are we to judge other people? Where does that where does? What does our faith, our Imam, our religion, tell us about that?P5

As a consequence, carers themselves may choose to keep their family member’s dementia secret:

It’s not something that some people don’t even know what it is, or what the you know, how it unfolds, or, you know, the different symptoms if you like, you know, some people don’t even know know these things. So the answer to your question is that I don’t, we didn’t have any support from the Asian community. Because they didn’t know. You know, we never told them. Okay.P5

#### 4.2.4. Response of Religious Leadership

If the lack of support from the wider Muslim community of British Pakistani origin is at least partly due to the need for more education and direction in their response to dementia, it follows that appropriate leadership may provide the answer. Here the picture is distinctly mixed. In the main, community leaders did not engage with the subject:

No, there’s no conversations [with religious or family leaders], I think, again, people take it for granted that, you know, as a Muslim, that’s my duty. So you wouldn’t really need to go to speak to somebody… I came to avoid using the word like, leaders, because leaders tend to be self-interested, often, their opinions, their perspective, they do not necessarily know how you’re doing and how you’re coping.P7

You know, so my, my caring from mum didn’t weaken my Islam, but my faith in Muslims, certain people who are meant to be important gatekeepers disappoint me, to be honest with you, because I feel they could do more, but they don’t.P5

However, there were two interviewees who reported that their Imam was addressing dementia in the community more directly:

I spoke with *Mufti* [an Imam], because he happens to be the next door neighbour so she could hear my mom being aggressive and violent. … the mufti said [Name], if you guys get unwell, who is going to care for your mum, so you are permitted to seek expert help support.P5

We see more and more Muslim elderly living on their own, the children move away, because of economic reasons or whatever. But the Imams I’ve spoken with, said that if it gets to the point where you are struggling as carers and family carers … then that’s permissible [seeking outside help] … The imam mentioned this in his *Khutbah* [Sermon] last Friday actually and he used the word dementia directly, which was ground-breaking, I think for a mufti to do this and in the khutbah said that you are permitted [to seek outside help], according to Islam.P5

This gives hope that, over time, increasing engagement by Imams will lead to an increased level of awareness and understanding in the community and a more nuanced understanding of a faithful Islamic response. 

To summarise the findings in this section, at present it seems that the wider Muslim community of British Pakistani origin is not engaging with questions of dementia or providing much active support to assist carers of family members living well with it. The underlying causes appear to be, first, a failure to separate the demands of Islam (e.g., looking after one’s parents) from cultural assumptions (e.g., that the best sort of care is always provided by the family, at home) so that the needs of carers are not taken into account, and may even be suppressed; and secondly, that the wider lack of awareness and understanding of dementia in the community can lead to apathy, stigma and secrecy. Most often, Imams and community leaders shared the same attitudes as the rest of the community; but in a few notable cases, Imams showed a good degree of awareness and were able to provide teaching on the subject of dementia that had the potential to change attitudes and be a source of support. This opens the possibility of enlisting the wider community in the support of people living with dementia and their carers by a programme of teaching and education.

### 4.3. Barriers to Accessing Services

As noted in the introduction, people living with dementia from communities of South Asian origin living in Britain tend to access dementia support services more rarely than their White British counterparts, and relatively late in the disease trajectory. Some contributing factors have already been identified, in the low levels of overall knowledge and understanding of dementia within the community and the high religio-cultural value given to care at home provided by a family member. However, additional barriers to access can be identified in the attitudes of Muslim carers of British Pakistani origin to these services, and the character of the services being offered to them.

#### 4.3.1. Knowledge, Attitudes and Understanding of the Community about Service Provision

As mentioned, there is cultural pressure on carers not to seek support from outside the family. This disincentive is compounded by ignorance of and anxiety about the available services. Participants described themselves as relatively ignorant of the services available to support their efforts to care for the family member living with dementia. They also expressed some concern that the help offered would be religiously and culturally inappropriate:

I wasn’t aware of any organizations that could help. GP is always the first people that we go to, or we go to an imam or a faith leader as Muslims. What happened was, she was assigned to a partner GP.P5

I don’t think there’s anything out there that is that Islamically that there’s a lot in the sense of materials to read, to help you with building your faith but actually anything practical, there’s nothing there. There’s no group that I could go and they’re all Muslim, [and who] would probably understand … There’s nowhere where you can go and have that you could offload or build that support network where you could, someone sharing that same experience so it reduces the isolation.P2

#### 4.3.2. Perceived Attitudes and Understanding of Service Providers towards the Community-Specific Dementia Care Concerns and Needs

If there is hesitancy from carers, participants also reported that it can be difficult to find service providers that can meet cultural and religious needs around dementia care:

And we couldn’t find anywhere in the locality that would meet our cultural, linguistic, religious, and dietary needs. So we found one eventually.P5

In addition, service providers themselves may assume that the majority of care will be provided by family members:

So if you, for example, when one couldn’t manage. … for mum the carer said that for me, as a daughter in law, as a Muslim, as a Pakistani [inaudible] my job is to just get on with it, and try and do the best that I can to support her, rather than try to find a solution out of the house.P7

## 5. Discussion and Conclusions

This study identified factors affecting access to dementia care that are consistent with the existing literature on communities of South Asian origin generally. However, it has also identified factors that are distinct to the religious beliefs and practices of the Muslim individuals and how these influence dementia care positively and/or negatively.

The data should be interpreted with caution, because of the limitations of the study. The sample size was unavoidably small, and the methodology of interviews by remote means (online and by telephone) may have limited the freedom with which participants were able to explore their thoughts and feelings when compared with face-to-face interviewing. The use of a single interview, conducted under COVID-19 restrictions without insight into the social or familial context, provides a ‘thin description’ of the current state of affairs that needs to be enriched with a ‘thicker’, extended and ethnographically informed programme of research [51]. In addition, the decision to interview only in English and by remote means may have restricted our access to carers who were older, more recently arrived in the UK or less integrated into society outside the British Pakistani community: it is noteworthy that only one participant was caring for a spouse (as opposed to a parent or grandparent) and none were over the age of 60. 

Despite these limitations, this initial study yielded some suggestive insights. In general, we found that family carers of people living with dementia in this sample group do draw strength and support from their faith at an individual level. It would be easy to overinterpret the data from a rather small sample, but reviewing the interview transcripts it is possible to discern how this may be taking place. Structured religious practices structure the interviewees’ lives and social relations (Section 4.1.1 above, and in this way enhance personal resilience and the acquisition of traits such as patience that enable family care (Section 4.1.2). We anticipated that, in turn, the burden of caring would interfere with and possibly undermine the religious practice of the carers, but this was not uniformly the case: on the contrary, practical difficulties were approached pragmatically and several interviewees spoke of the encouragement and support they received from the family member living with dementia (Section 4.1.3).

This positive picture did not extend to relations with the wider Muslim community, where respondents talked of confusion between properly religious and simply cultural expectations (Section 4.2.1); of a lack of awareness and understanding of dementia (Section 4.2.2) leading to a negative experience of both community (Section 4.2.3) and leadership (Section 4.2.4). Furthermore, carers may be discouraged from seeking support from service providers because of the cultural expectation of family-based care, concerns that the religious and cultural needs of the person living with dementia will not be understood or respected, and poorly informed responses from service providers themselves (Section 4.3).

There were however signs of positive change. One key finding that emerged clearly from this research is that most of these difficulties are not entrenched and structural ones but arise from unpreparedness or ignorance that can be relatively easily addressed. Thus, some carers spoke of knowing nothing about dementia at first, but quickly learning. Others were making pragmatic and constructive adjustments to fulfilling their Islamic duty for the family member living with dementia, involving critical distinctions between traditional cultural expectations and religious requirements within the cultural context of the UK. At the community level, there was evidence that dementia awareness among leaders can make a difference and be a source of support. For example, in two cases, religious leaders were found to be addressing the deficit of understanding of dementia within the community and providing advice on how to respond in an Islamic way. One Imam’s contribution probably proved decisive in helping one family to accept that the best way to care for their parent was to ask professionals to do it; and some service providers had clearly learned how to engage with the family in a responsive manner. 

These findings suggest that there is scope to improve the service to Muslims of British Pakistani origin living with dementia and their family carers by better-informed engagement with this community at the level of their beliefs, values and practices. Carefully planned, engaged, well-targeted training of both carers and service providers could yield significant improvements in the pattern of access to support services from this community. 

Consequently, to increase engagement between service providers and carers there is potential for development on three fronts. First, engagement with the community itself, where the active debate on the distinction between religious expectations and cultural ones suggests there is room for development and change with the help of an engaged and sensitive programme of education by working with sympathetic religious leaders. Secondly, there has been little thought or study devoted to the religious lives of carers, which they all claimed as a source of motivation strength and support: although the lazy notion that “They look after their own” has been used as an excuse not to engage with carers it remains the case that this is the context for most care. Taking the religious dimension of carers’ role seriously may help to establish lines of support that will enable them to fulfil their role more effectively and for longer, whilst their own needs as carers are also not neglected. Finally, service providers need to engage Muslim communities both to learn from them where the religiously based anxieties and concerns lie, and to reassure them of the commitment to religiously sensitive support. 

## Figures and Tables

**Table 1 healthcare-11-00120-t001:** Categories and Themes.

Category	Theme
Carers’ Religious Belief and Practice	Providing motivation to care
	Providing strength and support in caring
	Influenced by the experience of caring
Muslim community response to dementia	Culture/religion distinction
	Awareness and understanding of dementia
	Community as a barrier or potential source of support
	Response of religious leadership
Barriers to accessing services	Knowledge, attitudes and understanding of the community about service provision
	Attitudes towards and understanding of service providers

## Data Availability

Transcripts of the interviews can be examined by application to the Corresponding Author.

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
