# Peer review of "Religious Influences on the Experience of Family Carers of People with Dementia in a British Pakistani Muslim Community"

_healthcare, 2022, doi:10.3390/healthcare11010120_

Round 1

Reviewer 1 Report

The study is well thought out. Acknowledging the limitations that the study had provides room for continued research and study of the topic. This may also draw others to want to do a follow-up on the initial work. 

Author Response

Thanks you for your kind response. We have checked the English and corrected some errors in the format. These are recorded as tracked changes in the uploaded draft.

Reviewer 2 Report

Dear editors, dear authors, The paper is well structured, provides a good literature review that clearly identifies the gap in research that the paper seeks to address. As the authors acknowledge, the sample size is very small, but the aim of providing a scoping study, and not claiming data saturation, means that the paper is not seeking to pretend to offer something that it cannot deliver but instead offers a first impulse for future research. The small sample size may also have to do with the hard to reach target group, where research participation requires a significant amount of trust for "getting in", and in such underresearched settings even a small sample can in this context give valuable new insights. As such, the paper deserves to be published. A deficit of the research method is, in my opinion, the choice of online interviews (rather than in person) but this can be and is justified with reference to the exceptional circumstances of the Covid19 pandemic. The diversity of the team of authors is exemplary. With regards to the production of results, the team members first coded the interviews independently and developed possible categories for analysis, and then reflected jointly. This is one of the areas where the diversity of the team, and the possibly different perspectives that they bring to the text, can bear fruit. In line 174, there is suddenly a shift from the analysis to participant quotations, which is not made immediately obvious by the formatting. This should be changed, and the quotations should be more clearly separated from each other. The same happens all through the paper. Also, with regards to the quotations, I would have liked to see more engagement with the content of what the participants say, as the author's engagement with these statements is rather generic. In the analysis, for example, I would have liked to see an engagement of the key religious elements the participant mention (for example, fasting, prayer, submission under God's will). Maybe, however, the editors decide that for the current scope of the study the level of engagement is sufficient. Also due to my background in Religious Studies, I liked the section on the religion/culture distinction – something that could be looked into in more detail in future publications. In line 416, the word after the full stop should start with a capital letter. The sentence in line 418 does not really make sense to me. In line 430, a space should be inserted after the comma. Lines 503-505 should probably be removed. The suggestions for development on three fronts are constructively bringing the paper's insights to practice and policy. In sum, the paper offers a broad discussion of dementia care in a specific religious and cultural setting. The paper acknowledges its own limitations and manages to draw valuable conclusions that can lay the basis for important future research as well as serve as a foundational document for lobbying for a practical improvement of supporting Pakistani Muslim carers of people with dementia.

Author Response

Thank you for your thoughtful and comprehensive response. We have addressed your comments as follows:

" In line 174, there is suddenly a shift from the analysis to participant quotations, which is not made immediately obvious by the formatting. This should be changed"  This is mainly an error in the formatting: the new headings for this section were immediately under the table and at the end of the previous page, so effectively disappeared. We have adjusted the formatting so that they appear on line 174. 

"quotations should be more clearly separated from each other. The same happens all through the paper" These have been indented and reformatted throughout to clarify their distinction from the main text.

"with regards to the quotations, I would have liked to see more engagement with the content of what the participants say, as the author's engagement with these statements is rather generic. In the analysis, for example, I would have liked to see an engagement of the key religious elements the participant mention (for example, fasting, prayer, submission under God's will)." We accept that there was something missing here and have added a further discussion of the religious structure in lines 512-531

"In line 416, the word after the full stop should start with a capital letter. The sentence in line 418 does not really make sense to me. In line 430, a space should be inserted after the comma. Lines 503-505 should probably be removed." Corrections made as specified

Reviewer 3 Report

A few words of explanation of the term 'revert' (l. 158) might be appropriate since such (unfamiliar to me) designation has some relevance to the self-understanding of one of the researcher.

Apart from that this is a very clear and informative piece. Thank you.

Author Response

Thank you for your review. We have, as specified, added a brief explanation of the term 'revert' in a footnote: This term is used by some who ‘officially’ became Muslims in later life to indicate a belief that everybody is created in a relationship with God and rediscovers, rather than ‘converts’, to that faith